

# A new denoising approach based on mode decomposition applied to the stock market time series: 2LE-CEEMDAN

Zinnet Duygu Akşehir and Erdal Kılıç

Department of Computer Engineering, Ondokuz Mayıs University Samsun, Samsun, Turkey

## ABSTRACT

Time series, including noise, non-linearity, and non-stationary properties, are frequently used in prediction problems. Due to these inherent characteristics of time series data, forecasting based on this data type is a highly challenging problem. In many studies within the literature, high-frequency components are commonly excluded from time series data. However, these high-frequency components can contain valuable information, and their removal may adversely impact the prediction performance of models. In this study, a novel method called Two-Level Entropy Ratio-Based Complete Ensemble Empirical Mode Decomposition with Adaptive Noise (2LE-CEEMDAN) is proposed for the first time to effectively denoise time series data. Financial time series with high noise levels are utilized to validate the effectiveness of the proposed method. The 2LE-CEEMDAN-LSTM-SVR model is introduced to predict the next day's closing value of stock market indices within the scope of financial time series. This model comprises two main components: denoising and forecasting. In the denoising section, the proposed 2LE-CEEMDAN method eliminates noise in financial time series, resulting in denoised intrinsic mode functions (IMFs). In the forecasting part, the next-day value of the indices is estimated by training on the denoised IMFs obtained. Two different artificial intelligence methods, Long Short-Term Memory (LSTM) and Support Vector Regression (SVR), are utilized during the training process. The IMF, characterized by more linear characteristics than the denoised IMFs, is trained using the SVR, while the others are trained using the LSTM method. The final prediction result of the 2LE-CEEMDAN-LSTM-SVR model is obtained by integrating the prediction results of each IMF. Experimental results demonstrate that the proposed 2LE-CEEMDAN denoising method positively influences the model's prediction performance, and the 2LE-CEEMDAN-LSTM-SVR model outperforms other prediction models in the existing literature.

## INTRODUCTION

The data, consisting of observation values sorted by time, is a crucial source of information and strategy in various fields. Predicting new trends, behaviors, or potentially hazardous events based on past observations is essential. Forecasts based on time series data are widely applied in many fields, such as transportation, environment, and finance, and have

Corresponding author
Zinnet Duygu Akşehir,
duygu.aksehir@bil.omu.edu.tr

recently been of great interest from researchers (*Liu et al., 2019*; *Samal, Babu & Das, 2021*). For instance, accurate weather forecasting prevents loss of life and property due to natural disasters such as floods and landslides. Likewise, precise forecasting of stock trends or prices allows investors to make informed decisions in their trading activities, thereby facilitating profitable outcomes.

Time series are frequently used in the literature to address prediction problems. However, the inherent high noise and nonlinear, non-stationary properties of time series pose challenges to accurate prediction. Prediction models, sensitive to noise, encounter issues such as overfitting, underfitting, and suboptimal performance when applied to noisy data (*Dastgerdi & Mercorelli, 2022*; *Song, Baek & Kim, 2021*). To address this challenge, adopting an appropriate approach for noise elimination in the data is imperative. A review of literature focusing on noise reduction in time series reveals studies that cover diverse domains, including time series related to air-pollution (*Samal, Babu & Das, 2021*), Total Column of Ozone (TCO) (*Mbatha & Bencherif, 2020*), and electricity load/price (*Yaslan & Bican, 2017*; *Liu et al., 2019*). Notably, a significant portion of these studies concentrates on financial time series (*Qiu, Wang & Zhou, 2020*; *Tang et al., 2021*; *Liu et al., 2022a*; *Rezaei, Faaljou & Mansourfar, 2021*; *Cao, Li & Li, 2019*; *Lv et al., 2022*; *Yong'an, Yan & Aasma, 2020*; *Bao, Yue & Rao, 2017*; *Liu et al., 2022b*; *Zhang et al., 2023*). Two primary reasons underlie this situation: firstly, investors seek to enhance profit expectations through stock forecasting. The second reason is that financial time series exhibit higher noise than other time series. This noise is attributed to the influence of various factors, including company policies, political events, investor expectations, the general economic situation, and the non-stationary, non-linear characteristics inherent in financial data. For these reasons, this subject captures the attention of scientists in the field.

## Related work

Upon reviewing studies focused on noise elimination in financial time series, it becomes evident that denoising approaches relying on auto-encoder, Fourier transform, wavelet transform, and signal decomposition methods are commonly favored. These approaches are chosen to enhance the prediction performance of models susceptible to noise.

The autoencoder-based noise reduction approaches (*Zhao & Yang, 2023*; *Roostaee & Abin, 2023*; *Rekha & Sabu, 2022*) are complex and require intensive computation. Additionally, they lead to the loss of some significant features during the data compression process. This situation negatively impacts the performance of the developed model. Due to all these reasons, this method is not widely preferred for denoising in financial time series.

In the literature, a Fourier transform-based denoising approach has been employed to eliminate noisy components in stock market index data, particularly those that adversely impact the model's performance in predicting closing prices for the S&P500, KOSPI, and SSE indices (*Song, Baek & Kim, 2021*). In addition, the Fourier transform has also been used to eliminate the noise in the signals collected from the sensors for the damage detection module (*Yang et al., 2021*). In these studies, various deep-learning approaches

were employed to develop models on denoised data. The experimental results indicated that hybrid models outperformed basic models by combining denoising techniques with deep learning methods. The Fourier transform effectively divides a time series into frequency components and is particularly useful for identifying periodic patterns in the data, aiding in the analysis of continuous-time signals. However, it has limitations in describing time and frequency scale changes in time series data, making it less effective in analyzing time-varying signals (*Fourier, 1888*). Given its capability to overcome the limitations of the Fourier transform and its significant achievements in signal processing, the wavelet transform is employed in analyzing financial time series (*Qiu, Wang & Zhou, 2020*).

The wavelet transform-based denoising approach has been applied to eliminate noisy components in different stock market data, and subsequently, LSTM models were developed on the resulting noiseless data (*Dastgerdi & Mercorelli, 2022*; *Tang et al., 2021*; *Bao, Yue & Rao, 2017*). A hybrid noise reduction approach, combining the wavelet transform with complete ensemble empirical mode decomposition with adaptive noise (CEEMDAN) methods, was proposed to enhance further noise reduction in financial time series data (*Qi, Ren & Su, 2023*). While experimental results in these studies indicate a significant improvement in prediction stability through noise reduction using the wavelet transform, the method has some limitations. These limitations are that the effectiveness of Wavelet Transform in noise reduction depends on the number of decomposition layers and the choice of the basic wavelet function. However, the method's applicability is restricted, and the effectiveness of noise reduction is limited due to the absence of information about the appropriate values for parameters such as the number of decomposition layers and the choice of the basic wavelet function for time series data (*Liu et al., 2022a*). Denoising methods based on mode decomposition have gained prominence to address these drawbacks of the wavelet transform.

In the literature, various approaches that decompose time series into different frequency spectra, such as empirical mode decomposition (EMD) (*Zhang et al., 2023*; *Rezaei, Faaljou & Mansourfar, 2021*; *Lv et al., 2022*), ensemble empirical mode decomposition (EEMD) (*Wu & Huang, 2009*), variational mode decomposition (VMD) (*Wang, Cheng & Dong, 2023*; *Cui et al., 2023*), complete ensemble empirical mode decomposition (CEEMD) (*Rezaei, Faaljou & Mansourfar, 2021*; *Yong'an, Yan & Aasma, 2020*; *Liu et al., 2022b*), and CEEMDAN (*Cao, Li & Li, 2019*; *Lv et al., 2022*), are frequently preferred for denoising in time series. Examining studies in the literature reveals that the utilization of these approaches, combined with deep learning methods like long short-term memory (LSTM) and convolutional neural network (CNN), can mitigate the limitations of basic/single models (*Rezaei, Faaljou & Mansourfar, 2021*; *Cao, Li & Li, 2019*; *Lv et al., 2022*; *Yong'an, Yan & Aasma, 2020*; *Liu et al., 2022b*). Notably, results obtained with hybrid models, incorporating mode decomposition techniques and deep learning methods, demonstrate the superior performance of CEEMDAN-based models over others (*Cao, Li & Li, 2019*; *Lv et al., 2022*). When examining the hybrid methods based on mode decomposition applied

in the literature to predict stock market closing values, these methods were commonly implemented on indices such as S&P500, HSI, DAX, SSE, and DJIA. Additionally, it is noteworthy that LSTM, a popular deep learning method, is frequently the method of choice in these applications (*Rezaei, Faaljou & Mansourfar, 2021*; *Cao, Li & Li, 2019*; *Yong'an, Yan & Aasma, 2020*). However, in some studies, auto-regressive moving average (ARMA) (*Lv et al., 2022*) and feed-forward neural networks (FFNN) (*Liu et al., 2022b*) models are utilized to capture linear components resulting from decomposition. In all these studies, it is seen that the CEEMDAN method is preferred due to its advantages. The advantages of the CEEMDAN method to other mode decomposition methods can be listed as follows (*Torres et al., 2011*):

- *Adaptive noise control:* It is a technique that allows the adjustment of the amount of noise within a signal based on the characteristics and variability of the signal. The CEEMDAN method aims to achieve better results in noisy data by using adaptive noise control. In each decomposition stage of the signal, CEEMDAN analyzes the level of the noise component and adjusts the noise amount accordingly. This way, the optimal amount of noise for denoising the signal is determined, and the noise level is adjusted accordingly at each signal stage. As a result, the CEEMDAN method consistently achieves superior results compared to other methods, such as EMD, EEMD, and CEEMD, owing to its effective use of adaptive noise control.
- *Improved decomposition:* The CEEMDAN method prevents the occurrence of mode mixing, which is a problem in the EMD method where different modes interfere with each other. Mode mixing refers to similar oscillations in different modes or significantly different amplitudes in a single mode. The CEEMDAN method eliminates mode mixing by utilizing adaptive noise control and performing ensemble processing, which involves analyzing the data multiple times. By utilizing these techniques, the CEEMDAN method eliminates mode mixing more effectively than other methods, ensuring a more accurate and reliable signal decomposition.
- *Improved signal-to-noise ratio:* It provides a better signal-to-noise ratio through adaptive noise control.

This study introduced a two-level denoising approach named 2LE-CEEMDAN, incorporating entropy and CEEMDAN to remove noise from time series data effectively. This novel approach was tested on financial time series data, specifically stock market index data. In the initial step, the denoising technique was applied to the closing prices of stock market indices to eliminate noise from the data. Then, LSTM and SVR methods were applied to the denoised data to estimate the next day's closing values of the index. This methodology reflects a comprehensive strategy combining denoising and predictive modeling to enhance the accuracy of forecasting financial time series.

## Motivation and contributions

In recent years, there has been an increase in research efforts aimed at reducing noise in financial time series data. It has been observed that prediction models created using

denoising approaches proposed in these studies achieve more successful forecasting results than basic models. Based on the mode decomposition in the literature, it is evident that removing high-frequency components in methods developing trading strategies results in information loss, adversely affecting the model performances. Addressing how to mitigate this information loss is an open problem.

The primary motivation of this study is to develop a new approach that effectively separates noisy data from financial time series without causing any loss of useful information. Therefore, this aspect distinguishes this study from existing literature, emphasizing a novel perspective.

The fundamental contributions of the study are provided below:

- A novel denoising method, named 2LE-CEEMDAN, has been developed based on CEEMDAN to extract valuable information from high-frequency components discarded as noise.
- A new methodology has been proposed to effectively identify noisy components from the IMFs obtained through the decomposition of time series data by utilizing approximate and sample entropy to measure irregularities in the time series data.
- To demonstrate the method's effectiveness, a new hybrid prediction model has been presented for predicting the closing prices of stock market indices. This hybrid model includes the 2LE-CEEMDAN denoising approach with LSTM and SVR methods.

### Organization

The rest of this study is organized as follows: In the 'Methodology' section, the general framework of the proposed prediction model within the scope of the study is given, and the approaches used in this method are detailed. The 'Experimental Settings' section mentions the used dataset for the prediction model, how the hyperparameter adjustments of the model are made, and performance evaluation metrics. The results obtained in the 'Results and Discussion' section is interpreted by giving them in tables and figures. The results are discussed in the 'Conclusion and Future Works' section, and future works are evaluated.

## METHODOLOGY

In this section, firstly the general framework of the 2LE-CEEMDAN-LSTM-SVR prediction model and the essential stages of the architecture are explained. Secondly, the 2LE-CEEMDAN denoising approach used in this architecture is explained. Finally, the forecasting model details are given.

### Framework

The problem under investigation in this study is the examination of the impact of noise on a predictive model designed to forecast the price of any commodity. To demonstrate this impact and predict the closing value of the next day's stock market index, the study proposes a forecasting model called 2LE-CEEMDAN-LSTM-SVR. This comprehensive model has five stages, consisting of two level-based CEEMDAN, entropies, LSTM, and

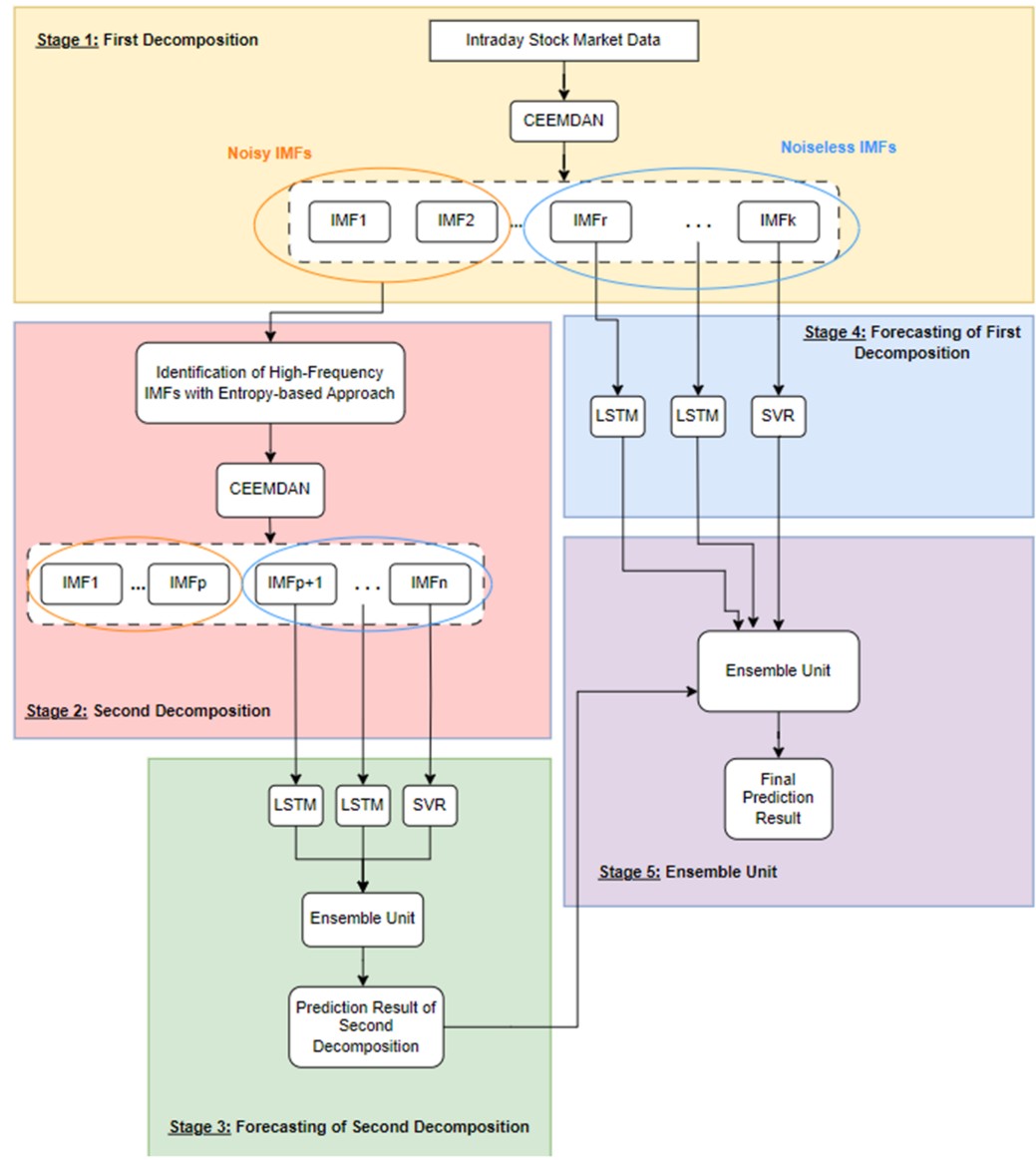

**Figure 1** The framework of the proposed forecasting model.

SVR. The framework of the model is visually represented in Fig. 1, with detailed explanations provided below:

- **Stage 1-First decomposition:** The time series data is decomposed using the CEEMDAN method at this stage. Subsequently, entropy ratios are computed using the approximate and sample entropy values for each IMF obtained from the decomposition process. IMFs with ratios surpassing a predefined threshold value for entropy ratios are classified as high-frequency components, indicating noisy IMF components. Conversely, those with ratios below the threshold are identified as noiseless IMFs.

- **Stage 2-Second decomposition:** In this stage, which also marks the second step of the 2LE-CEEMDAN method, a different approach is taken compared to directly discarding the IMFs identified as high-frequency in Stage 1. In contrast to directly discarding the IMFs identified as high-frequency in Stage 1, the second decomposition process is applied to these data. This is done because these IMFs may contain valuable information. The procedures outlined in Stage 1 are replicated similarly to categorize the obtained IMFs as either noisy or noiseless. IMFs determined to be noiseless are selected to be used in the training phase.

- **Stage 3-Forecasting of second decomposition:** The noiseless components identified in Stage 2 undergo separate training using LSTM and SVR methods. The final IMF obtained from the decomposition, often referred to as the residual, is characterized by linear properties and is therefore trained using the SVR model. Conversely, the remaining noiseless IMFs exhibit nonlinear properties and are trained using the LSTM model. The prediction results from each IMF are subsequently combined, providing a comprehensive and integrated forecasting outcome that leverages the strengths of both linear and nonlinear modeling approaches. This multi-model strategy aims to capture and utilize the distinctive features of each component for enhanced predictive accuracy.

- **Stage 4-Forecasting of first decomposition:** The residue component determined in Stage-1 is trained with the SVR method, and the remaining IMFs are trained with the LSTM method.

- **Stage 5-Ensemble unit:** It is a simple ensemble unit where the average is calculated after the prediction results from Stages 3 and 4 are summed. Thus, the final prediction result was obtained.

## 2LE-CEEMDAN denoising method

To comprehend the foundation of the 2LE-CEEMDAN denoising method, it is essential to first detail the explanations of the CEEMDAN and entropy concepts.

### CEEMDAN

The EMD method, introduced by *Huang et al. (1998)*, divides a time series into fundamental components known as IMFs. This method can decompose complex nonlinear and non-stationary time series data into multiple IMF components, with the last IMF referred to as the residue. These obtained IMFs are used in data analysis or understanding the internal structure of a time series. However, the practical implementation of EMD often leads to the issue of mode mixing, characterized by very similar oscillations in different modes or significant differences in amplitudes within a single mode. To address the mode mixing problem, the EEMD was proposed as an enhanced version by *Wu & Huang (2009)*. EEMD defines IMF components by averaging over multiple trials, introducing white noise to each trial to achieve less noisy IMFs than EMD. While EEMD overcomes the mode mixing problem and provides less noisy IMFs, it has a high computational cost and cannot eliminate white noise during signal reconstruction. To address these limitations, *Torres et al. (2011)* introduced the

---

**Algorithm 1 CEEMDAN decomposition method.**

Output: $\widetilde{IMF}_k$

$r_k(n) \leftarrow residues$

$E_j(.) \leftarrow j$-th IMF obtained by EMD decomposition

$w^i \leftarrow$ White noise

$x(n) \leftarrow$ Time series signal

$\varepsilon_0 \leftarrow$ Noise coefficient

$I \leftarrow$ Number of trials

$IMF_1^i(n) = x(n) + \varepsilon_0 w^i(n)$

$\widetilde{IMF}_1(n) = 0$

**for** $i = 1$ to $I$ **do**

   $\widetilde{IMF}_1(n) = \widetilde{IMF}_1(n) + IMF_1^i(n)/I \leftarrow$ First IM

**end for**

$r_1(n) = x(n) - \widetilde{IMF}_1(n) \leftarrow$ First residue

$\widetilde{IMF}_2(n) = 0$

**for** $i = 1$ to $I$ **do**

   $\widetilde{IMF}_2(n) = \widetilde{IMF}_2(n) + E_1(r_1(n) + \varepsilon_1 E_1(w^i(n)))/I \leftarrow$ Second IMF

**end for**

**while** $r_k(n) \leftarrow$ until the value of residual component is less than two extremes **do**

   **for** $k = 2$ to K **do**

      $r_k(n) = r_{k-1}(n) - \widetilde{IMF}_k(n) \leftarrow$ Residuals for $k = 1, 2, ..K$

      $\widetilde{IMF}_k(n) = 0$

      **for** $i = 1$ to $I$ **do**

         $\widetilde{IMF}_{k+1}(n) = \widetilde{IMF}_{k+1}(n) + E_1(r_k(n) + \varepsilon_k E_k(w^i(n)))/I$

      **end for**

   **end for**

**end while**

---

CEEMDAN as an improved version of EEMD. CEEMDAN stands out among mode decomposition methods due to its effective elimination of mode mixing, almost zero reconstruction error, and significantly reduced computational cost. When given a non-stationary and nonlinear signal $X(t)$, the decomposition is performed using the CEEMDAN method, following the steps outlined in Algorithm 1.

### Entropy

Entropy is a concept that quantifies the complexity or irregularity of a time series. A higher entropy value indicates greater complexity and irregularity in the time series. In financial time series, a high entropy value implies speculative price movements. In the literature, two prominent entropy approaches based on information theory are approximate entropy and

---

**Algorithm 2** Calculate the approximate and sample entropy.

Input: *Timeseries*

Output: *ApEn, SampEn*

$x(l), l = 1, 2, .., M \leftarrow$ Given a time series

$p \leftarrow$ Embedding dimension

$z \leftarrow$ Tolerance

**Stage 1:** Extend *x(l)* to the $p^{th}$ vector $U_p(l)$

$U_p(l) = [x(l), u(l+1), ..., u(l+p-1)] \leftarrow l = 1, 2, ..., M - p + 1$

**Stage 2:** Calculate the distance between $U_p(l)$ and $U_p(j)$

$D[U_p(l), U_p(j)] = max_{t=0,1,...,p-1}\{|x(l+t) - x(j+t)|\} \leftarrow j = 1, 2, ..., M - p + 1$ and $j \neq l$

**Stage 3:** Compute approximate entropy

- *Measure the regularity and frequency of patterns within tolerance r:*

$$C_l^p(z) = \frac{\text{Number of j such that } D[U_p(l), U_p(j)] \leq z}{M - p + 1}$$

- Compute the mean value of the logarithm of $C_l^p(z)$:

$$\psi^p(z) = \frac{\sum_{l=1}^{M-p+1} ln[C_l^p(z)]}{M - p + 1}$$

- The ApEn can be defined:

$$ApEn(p, z) = \psi^p(z) - \psi^{p+1}(z)$$

**Stage 4:** Compute sample entropy

- Compute the two coefficients:

$$A_l^p(z) = \frac{\sum_{j=1,j\neq l}^{M-p} \text{number of times that } D[U_{p+1}(l), U_{p+1}(j)] < z}{M - p - 1}$$

$$B_l^p(z) = \frac{\sum_{j=1,j\neq l}^{M-p} \text{number of times that } D[U_p(l), U_p(j)] < z}{M - p - 1}$$

- Add them as fallow:

$$A^p(z) = \frac{\sum_{l=1}^{M-p} A_l^p(z)}{M - p}$$

*(Continued)*

**Algorithm 2** (**continued**)

$$B^p(z) = \frac{\sum_{l=1}^{M-p} B_l^p(z)}{M-p}$$

- The SampEn can be defined:

$$SampEn(p, z) = -ln\left[\frac{A^p(z)}{B^p(z)}\right]$$

---

**Algorithm 3** **2LE-CEEMDAN denoising method.**

Input: *Timeseriesdata*

$IMFs : [IMF_0, IMF_1, ..., IMF_{k-1}] \leftarrow$ k IMFs obtained with CEEMDAN() (Algorithm 1)

**procedure** SELECTION_IMF(*IMFs*)

    *sample_entropy_list* = []

    *approximate_entropy_list* = []

    **for** imf **in** IMFs **do**

        *sample_entropy_list.append(SampEn(imf))*

        *approximate_entropy_list.append(ApEn(imf))*

        *total_SampEn* = *sum(sample_entropy_list)*

        *total_ApEn* = *sum(approximate_entropy_list)*

        *sample_entropy_ratio* = $\left[\frac{entropy*100}{total\_SampEn}\right.$ for entropy in sample $_e$ntropy list $\left.\right]$

        *approximate_entropy_ratio* = $\left[\frac{entropy*100}{total\_ApEn}\right.$ for entropy in approximate entropy list $\left.\right]$

    **end for**

    *noisy_imfs* = []

    *noiseless_imfs* = []

    **for** i **in** range(len(IMFs)) **do**

        **if** *sample_entropy_ratio[i]* > 20 || *approximate_entropy_ratio[i]* > 20 **then**

        *noisy_imfs.append(IMFs[i])*

        **else**

        *noiseless_imfs.append(IMFs[i])*

        **end if**

    **end for**

    **return** *noisy_imfs, noiseless_imfs*

**end procedure**

*First_IMFs* = *CEEMDAN(Input)*

*First_decomposition_noisy_IMF, First_decomposition_noiseless_IMF* = *Selection_IMF(First_IMFs)*

---

| Algorithm 3 (continued) |
| --- |
| $noisly\_First\_IMFs\_sum = sum(First\_decomposition\_noisy\_IMF)$ |
| $Second\_IMFs = CEEMDAN(noisly\_First\_IMFs\_sum)$ |
| $Second\_decomposition\_noisy\_IMF, Second\_decomposition\_noiseless\_IMF = Selection\_IMF(Second\_IMFs)$ |

sample entropy. These entropy metrics provide a mathematical measure of the amount of information, with theoretical distinctions between them (*Pincus, 1991*; *Richman & Moorman, 2000*). Approximate entropy (*ApEn*) and sample entropy (*SampEn*) can mitigate noise-induced complexity in time series. Establishing a threshold value for noise in these criteria proves sufficient for this purpose. Numerous experiments have indicated that the proportional use of these two metrics positively contributes to transaction accuracy. This approach systematically facilitates the correct identification of IMFs containing noisy components. Algorithm 2 outlines the calculation of *ApEn* and *SampEn* values for the IMFs obtained from the decomposition of time series using CEEMDAN in the proposed method.

### 2LE-CEEMDAN

The 2LE-CEEMDAN method, which we propose to effectively eliminate noise in time series, performs two-level decomposition. While applying this method, the steps in Algorithm 3 are followed:

1) The initial step involves decomposing the time series using the CEEMDAN method as outlined in Algorithm 1, resulting in the acquisition of IMFs.

2) For each obtained IMF, the approximate and sample entropy values are initially computed using Algorithm 2. Subsequently, Algorithm 3 is employed to sum the entropy values of all IMFs for each entropy metric. This procedure results in the determination of *total_SampEn* and *total_ApEn*. Finally, *approximate_entropy_ratio* and *sample_entropy_ratio* values are computed.

3) The noisy-noiseless components are identified by calculating the entropy ratios for each IMF. Accordingly, IMFs with a ratio above the predetermined common threshold for entropy ratios are called high-frequency, in other words, noisy components, while the others are called noiseless. Since the possibility that these noisy components contain valuable information, as opposed to directly discarding them, these components are collected and performed a second decomposition with the assistance of Algorithm 1.

4) For IMFs obtained as a result of the second decomposition process, step 2 is performed similarly. Then the noisy and noiseless components are determined, and the algorithm is terminated.

The output of this method is the noiseless IMFs obtained as a result of the first and second decomposition.

## Forecasting model

Following the successful removal of noisy components from the time series data through the 2LE-CEEMDAN algorithm, the noiseless IMFs obtained from the first and second decomposition stages constitute the input data for the forecasting model. This process is explained in Stages 3–5, as depicted in the framework presented in Fig. 1. IMFs characterized by nonlinear attributes are fed into the LSTM model. The last IMF (residue) with linear characteristics derived from the decomposition process is also given as an input to the SVR model. After separately were trained each IMF, the prediction results were ensembled. In other words, as shown in Fig. 1, the prediction results in Stage 3 and 4 are combined hierarchically to obtain the final prediction result.

### LSTM

Unlike traditional neural networks, recurrent neural networks (RNN) have loops in their architecture and are based on the logic of using sequential information (*Elman, 1990*). RNN architectures are widely used in many applications, such as translation, voice recognition, language modeling, and handwriting recognition because they can establish and interpret relationships between sequential data (*Mikolov et al., 2010*). But when implementing these applications with RNN architectures, two main problems arise:

- *Long dependency problem:* RNN networks can make sense of and interpret data by connecting with the past, thanks to their architecture. But when it goes too far back, it cannot perform this process and establish the necessary connection with the past. In this case, the so-called long dependency problem arises (*Olah, 2015*).
- *Vanishing gradient problem:* One of the problems in neural networks that makes it difficult to update the weights of previous layers with backpropagation is the excessive number of layers. The partial derivative is used when calculating the weights of the previous layers in the backpropagation process. Gradient values are calculated with the help of this partial derivative. But when the number of layers increases, the gradient values will approach zero after a certain point in the backpropagation process, and the weights will not be updated over time. In this case, learning will also be difficult for the network. This problem is defined as the vanishing gradient problem (*Young et al., 2018*).

LSTM networks, a special structure of RNN, were introduced by *Hochreiter & Schmidhuber (1997)*, especially to solve the long dependency problem of RNN. LSTM also solves the vanishing gradient problem, another problem of RNN. Also, LSTM consists of modules that repeat each other, such as RNN. But when the architectures of the two networks are compared, the main difference is the number of layers. The RNN architecture has a single layer containing a tanh layer, while the LSTM architecture in Fig. 2 consists of four consecutive layers, unlike RNN.

LSTM uses these four layers to ensure that information is remembered by the network for long periods. The LSTM transaction equations are given in Eqs. (1)–(6):

$$f_t = \sigma(W_f \cdot [h_{t-1}, X_t] + b_f) \tag{1}$$

$$i_t = \sigma(W_i \cdot [h_{t-1}, X_t] + b_i) \tag{2}$$
$$o_t = \sigma(W_o \cdot [h_{t-1}, X_t] + b_o) \tag{3}$$
$$\widetilde{C}_t = tanh(W_c \cdot [h_{t-1}, X_t] + b_c) \tag{4}$$
$$C_t = f_t \odot C_{t-1} + i_t \odot \widetilde{C}_t \tag{5}$$
$$h_t = o_t \odot tanh(C_t) \tag{6}$$

From the above equations, for an input vector $X$ the LSTM unit at time step $t$: it is an input gate, $f_t$ is a forget gate, $o_t$ is an output gate, $C_t$ is a memory cell, $h_t$ is the hidden state, $W$ is weight matrix, $b$ is bias vector and $\sigma$ activation function. The default connections among these units are presented in Fig. 2.

### SVR

Support vector regression is a form of the support vector machines method developed by *Cortes & Vapnik (1995)*. SVR is an approach that divides data points by a hyperplane and creates a regression model on this plane. The hyperplane performs predictions by ensuring that as many data points as possible remain above it. The basic idea is to find a regression function that passes through two margins where the data points are divided by a hyperplane and estimate the value of the dependent variable through this function. In other words, SVR aims to reduce the error by minimizing the distance between the predicted and observed values by determining the hyperplane. A linear SVR tries to find a regression function expressed by $f(x, w, w_0) = w^T x + w_0$ in hyperspace. Here, $f(x, w, w_0)$ denotes the output value, $x$ denotes the input values, $w$ represents the linear weights, and $w_0$ denotes the correction term. In SVR, the aim is to find the function $f(x, w, w_0)$ by minimizing the regression risk given in Eq. (7).

$$min \frac{1}{2} \|w\|^2 + C \sum_t (\varepsilon_+^t + \varepsilon_-^t) \tag{7}$$

Here $C$ denotes an adjustment parameter controlling the error, and $\varepsilon_+^t$ and $\varepsilon_-^t$ denote the deviations from the plane, respectively. The constraints of this function are given in Eqs. (8)–(10).

$$y - f(x, w, w_0) \leq \varepsilon + \varepsilon_+^t \tag{8}$$
$$f(x, w, w_0) - y \leq \varepsilon + \varepsilon_-^t \tag{9}$$
$$\varepsilon_+^t, \varepsilon_-^t \geq 0 \tag{10}$$

From the above constraints, $y$ denotes the actual value of the dependent variable, and $\varepsilon$ denotes the error tolerance. The Lagrangian method is used for the solution of this optimization method, and as a result of the solution, the $f(x, w, w_0)$ function is obtained in Eq. (11). Where $\alpha_+^t$ and $\alpha_-^t$ are Lagrange multipliers, and $K$ denotes the kernel function.

$$f(x, w, w_0) = \sum_t (\alpha_+^t - \alpha_-^t) K(x^t, x) + w_0 \tag{11}$$

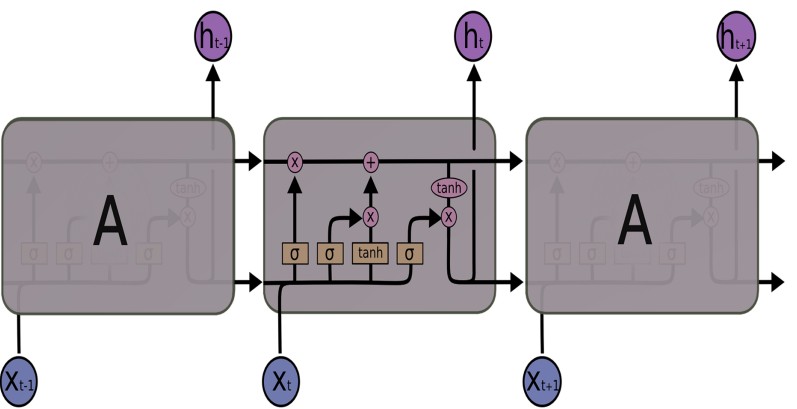

**Figure 2  LSTM architecture.**

The critical parameters for the SVR method are:

- *C:* It is a regularization parameter that controls the error. This parameter controls the model's fit to the data points. A small *C* value tolerates errors, while a greater *C* value attempts to minimize the errors. The *C* parameter can be adjusted to control the trade-off between overfitting and underfitting.
- *Epsilon (ε):* It determines the error tolerance. This parameter sets the maximum distance between the hyperplane and the data points. The epsilon value controls how much error is acceptable and how close the model should be to the data points.
- *Kernel function (K):* SVR can be applied to linear and non-linear datasets. In this regard, the kernel function captures linear or non-linear relationships. For example, a linear kernel is used for linear relationships, while an Radial Basis Function (RBF) kernel function can be preferred for capturing non-linear relationships. The choice of kernel function is made considering the dataset's structure and the prediction problem's nature

## EXPERIMENTAL SETTINGS

Various experiments were carried out to evaluate the effectiveness of the proposed 2LE-CEEMDAN denoising method in predicting the next day's closing value of stock market indices. The denoising method was applied to four stock market index datasets, eliminating noise and subsequently creating a prediction model using the obtained noiseless components. This section introduces the datasets utilized in the study and then outlines the construction of the forecasting model for these datasets. The conclusion of the section provides metrics related to evaluating the forecasting model's performance.

### Dataset

To assess the impact of the proposed 2LE-CEEMDAN denoising approach on the prediction model's performance for one-step-ahead prediction, the following four major global stock indices were selected:

- *Standard and Poor's 500 (S&P500):* It is a stock index comprising 500 prominent U.S. companies. It encompasses around 75% of the American stock market.
- *Shanghai Stock Exchange Composite (SSE):* It is a stock market index that reflects the performance of all A-shares and B-shares listed on the Shanghai Stock Exchange.
- *Deutscher Aktien (DAX):* It is a German stock market index representing the performance of the 30 largest and most liquid companies trading on the Frankfurt Stock Exchange. These companies are considered major players in the German economy and are carefully selected to provide a comprehensive overview of the country's stock market.
- *Dow Jones Industrial Average (DJI):* It is a stock market index that measures the performance of 30 large and well-established companies listed on stock exchanges in the United States. The DJI includes companies from various sectors, such as technology, finance, health care, and manufacturing.

Daily closing values for these stock indices were collected from Yahoo! Finance for the period between January 1, 2010, and October 1, 2019.

The statistical analysis results, including the amount of data, minimum, maximum, average, and standard deviation information for each stock market index, are presented in Table 1. The table shows a substantial range between the minimum and maximum values of the closing prices for the stock market indices, and the standard deviation values are notably high. This indicates that the selected stock market indices exhibit high volatility and possess a non-stationary property.

## Construction of the forecasting model

The 2LE-CEEMDAN-LSTM-SVR forecasting model, described in the "Forecasting Model" section, and designed to predict the next day's closing value of stock market indices, is constructed as follows:

- First, the 2LE-CEEMDAN method, given a pseudo-code in Algorithm 3, was applied to the time series data consisting of closing values for each stock market index. Here, two-level decomposition and then entropy ratio-based denoising was performed using the CEEMDAN method. For the CEEMDAN method used in the decomposition of the stock market index data, the number of trials and white noise standard deviation values were adjusted to 200 and 0.2, respectively, taking reference from the study of *Liu et al. (2022b)*. The values of the embedding dimension and tolerance parameters in approximate entropy and sample entropy calculated for each IMF were determined as 2 and 0.2, respectively, as a result of various experiments. In addition, the noise threshold value for the entropy ratio used to determine noisy and noiseless components was set as 20 (seen Algorithm 3). Therefore, the IMF components with approximate and sample entropy ratios above 20% were noisy, and the rest were noiseless.
- Following the successful elimination of noise from the stock index data, the noiseless IMFs obtained from the first and second decompositions serve as input features for both the LSTM and SVR models. Each IMF component, identified as noiseless through the

**Table 1 Descriptive statistics of stock indices closing price.**

| Index | Count | Mean | Max | Min | Standard deviation |
|-------|-------|------|-----|-----|--------------------|
| DAX | 2,470 | 9,445.98 | 13,559.60 | 5,072.33 | 2,350.25 |
| DJI | 2,452 | 17,347.54 | 27,359.16 | 9,686.48 | 4,951.69 |
| S&P500 | 2,452 | 1,933.37 | 3,025.86 | 1,022.58 | 567.36 |
| SSE | 2,366 | 2,795.73 | 5,166.35 | 1,950.01 | 537.10 |

1st and 2nd decompositions, undergoes separate training in this phase. The training process involves scaling the IMFs to the range (0,1) using the min-max scaler. Subsequently, the data is divided into training (90%) and test (10%) datasets. While the residual one of these IMFs is trained with SVR, the others are trained with the LSTM model. The final prediction result of the forecasting model is achieved by hierarchically combining the prediction results obtained from the individually trained IMFs, as depicted in Stages 3–5 of Fig. 1.

The hyperparameter settings for the LSTM and SVR prediction models are detailed in Table 2. The hyperparameter tuning process for the SVR method utilized a grid search approach. Hyperparameter values for LSTM were chosen based on relevant literature studies, incorporating commonly preferred values in this study. Experiments were executed by generating training-test sets concerning three different values for the time step parameter specified in the table. Moreover, to mitigate overfitting during training in the LSTM method, a dropout layer with a ratio of 0.1 was introduced between both hidden layers. Additionally, early stopping was implemented.

## Performance evaluation

We evaluated the proposed 2LE-CEEMDAN-LSTM-SVR forecasting model using the mean absolute percentage error (MAPE), mean absolute error (MAE), root mean square error (RMSE), and $R^2$, which are often used in the literature. The RMSE, MAE, MAPE, and $R^2$ metrics are calculated using the Eqs. (12)–(15), respectively. In the below equations, $n$ refers to the number of data, $P$ refers to the estimated value, $A$ refers to the actual value, and $\widetilde{A}$ refers to the average of actual values.

$$RMSE = \sqrt{\frac{1}{n}\sum_{i=1}^{n}(P_i - A_i)^2} \tag{12}$$

$$MAE = \frac{1}{n}\sum_{i=1}^{n}|P_i - A_i| \tag{13}$$

$$MAPE = \frac{1}{n}\sum_{i=1}^{n}\left|\frac{P_i - A_i}{A_i}\right| \tag{14}$$

$$R^2 = 1 - \frac{\sum_{i=1}^{n}(P_i - A_i)^2}{\sum_{i=1}^{n}(P_i - \widetilde{A_i})^2} \tag{15}$$

**Table 2 Hyperparameter settings of the forecasting model.**

| Method | Parameters | Values |
|---|---|---|
| LSTM | Time step | 5, 10, 15 |
| | Number of hidden layers | 3 |
| | Number of units | 128, 64, 16 |
| | Activation function | ReLU |
| | Optimizer | Adam |
| | Loss function | Mean square error |
| | Regularization | Early stopping |
| | Max number of epochs | 200 |
| SVR | Time step | 5, 10, 15 |
| | Kernel function | Linear |
| | Cost | 10 |
| | Epsilon | 0.001 |
| | Cross-validation | 5-fold |

# RESULTS AND DISCUSSION

This section discusses the findings obtained from experiments on DAX, DJI, SSE, and S&P500 indices to evaluate the forecasting performance of the proposed 2LE-CEEMDAN-LSTM-SVR forecasting model. The S&P500 stock market index was chosen as an illustrative case to demonstrate the noise elimination process in the time series.

Noise reduction was executed on this time series data by following the steps outlined in Algorithm 3. The closing values of the S&P500 and the resulting IMFs from the first decomposition are depicted in Fig. 3, while the calculated entropy values and ratios for the IMFs are presented in Table 3A. Considering the entropy ratios, the first three IMFs were identified as noisy, whereas the remaining IMFs were labeled noiseless. High-frequency data was obtained by collecting the noiseless components determined from the first decomposition. The high-frequency data and the IMFs from the second decomposition are shown in Fig. 4. Similarly, the entropy values and ratios for the IMFs obtained at the second decomposition stage are provided in Table 3B ($IMFi^{(j)}$ denotes the i-th IMF from the j-th decomposition.). Upon inspecting the entropy ratios of the IMFs, it was observed that the first 4 IMFs contained noise. Consequently, the first four noisy IMFs were discarded at this stage, and the noiseless IMFs resulting from the first and second decompositions were utilized for training in the forecasting phase. All these steps were replicated for the other three stock market indices.

To evaluate the effect of the 2LE-CEEMDAN denoising method on the forecasting model, we conducted the following scenarios:

- **Scenario-1:** Financial time series are decomposed using the CEEMDAN method. In this scenario, no denoising approach was applied; only decomposition was carried out on the financial time series. The last IMF, obtained from the decomposition, is trained with the SVR method, while the remaining IMFs are trained separately using the LSTM method.

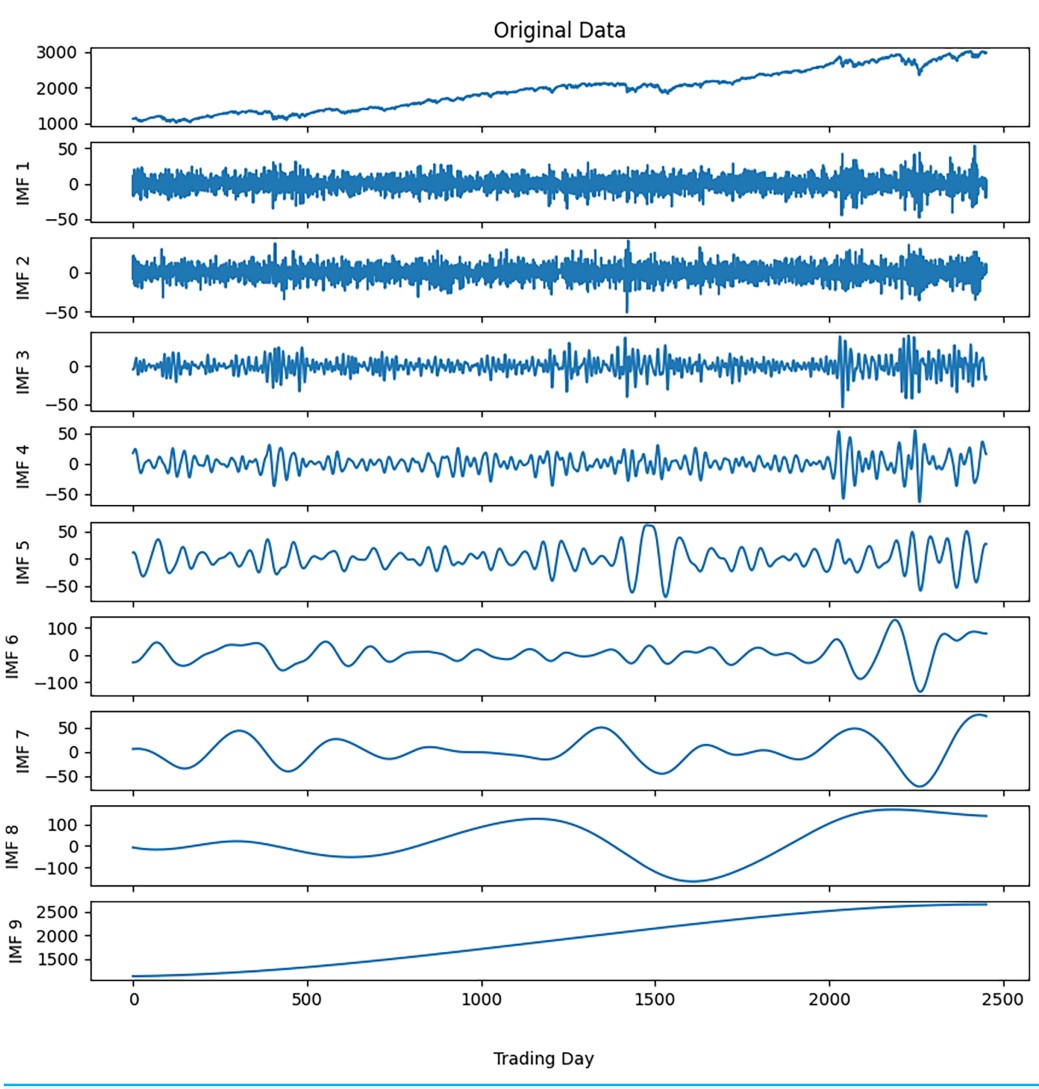

**Figure 3** S&P500 first decomposition results.

**Table 3** Entropy values and ratios of IMFs obtained from first and second decomposition for the S&P500.

| | (A) First decomposition | | | | | (B) Second decomposition | | | |
|---|---|---|---|---|---|---|---|---|---|
| | ApEn | SampEn | ApEn ratio | SampEn ratio | | ApEn | SampEn | ApEn ratio | SampEn ratio |
| $IMF1^{(1)}$ | 0.2035 | 4.3820 | 9.9552 | 31.9381 | $IMF1^{(2)}$ | 0.4619 | 3.3763 | 17.0960 | 38.6948 |
| $IMF2^{(1)}$ | 0.2161 | 4.4567 | 10.5730 | 32.4821 | $IMF2^{(2)}$ | 0.4676 | 2.7142 | 17.3073 | 31.1064 |
| $IMF3^{(1)}$ | 0.4251 | 2.1043 | 20.7939 | 15.3370 | $IMF3^{(2)}$ | 0.5578 | 1.4665 | 20.6438 | 16.8069 |
| $IMF4^{(1)}$ | 0.3518 | 1.0292 | 17.2095 | 7.5015 | $IMF4^{(2)}$ | 0.5956 | 0.6657 | 22.0416 | 7.6292 |
| $IMF5^{(1)}$ | 0.2572 | 0.6898 | 12.5807 | 5.0279 | $IMF5^{(2)}$ | 0.4655 | 0.3486 | 17.2281 | 3.9949 |
| $IMF6^{(1)}$ | 0.2077 | 0.5092 | 10.1626 | 3.7116 | $IMF6^{(2)}$ | 0.1164 | 0.1182 | 4.3076 | 1.3542 |
| $IMF7^{(1)}$ | 0.2915 | 0.3453 | 14.2621 | 2.5168 | $IMF7^{(2)}$ | 0.0277 | 0.0259 | 1.0267 | 0.2964 |
| $IMF8^{(1)}$ | 0.0896 | 0.1818 | 4.3813 | 1.3249 | $IMF8^{(2)}$ | 0.0068 | 0.0071 | 0.2498 | 0.0811 |
| $IMF9^{(1)}$ | 0.0017 | 0.0220 | 0.0817 | 0.1601 | $IMF9^{(2)}$ | 0.0027 | 0.0032 | 0.0992 | 0.0361 |

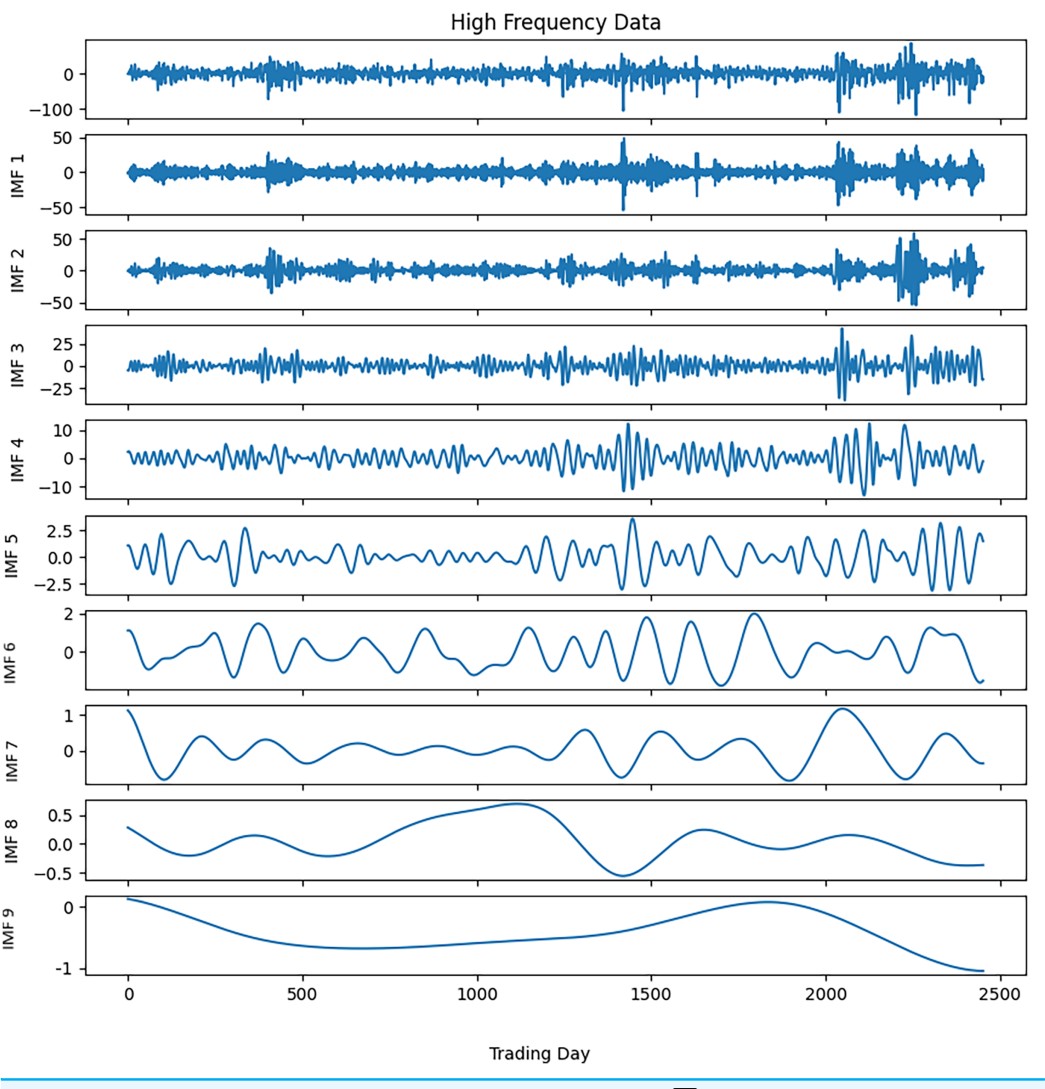

**Figure 4  S&P500 second decomposition results.**

The final prediction result is obtained by ensembling the prediction results of the individual IMFs.

- **Scenario-2:** After decomposing the financial time series with the CEEMDAN method, noisy and noiseless components were identified using the procedure outlined in Algorithm 3. In this scenario, the noisy IMFs were directly discarded, and the last of the remaining noiseless IMFs was trained separately with the SVR method, while the rest were trained with the LSTM method. The final prediction result was obtained by ensembling the prediction results of the individual IMFs.

- **Scenario-3:** In this scenario, proposed the 2LE-CEEMDAN-LSTM-SVR forecasting model was used.

Scenarios 1–3 were applied to predict the next day's closing value by considering the past 5, 10, and 15 observation values of the S&P500, DAX, DJI, and SSE stock indices. The estimation results obtained through these scenarios are given in Table 4 respectively.

| Table 4 Forecasting results in all scenarios for the indices. | | | | | | |
|---|---|---|---|---|---|---|
| Index | Scenario | Time step | RMSE | MAE | MAPE | $R^2$ |
| S&P500 | Scenario 1 | 5 | 6.9013 | 5.3612 | 4.9323 | 0.8046 |
| | | 10 | 7.4713 | 6.1221 | 5.4440 | 0.6666 |
| | | 15 | 7.3907 | 6.1057 | 5.0829 | 0.6903 |
| | Scenario 2 | 5 | 6.1225 | 4.7923 | 2.2288 | 0.9431 |
| | | 10 | 7.3718 | 6.2237 | 2.3221 | 0.7003 |
| | | 15 | 6.6946 | 5.8330 | 2.3448 | 0.8106 |
| | Scenario 3 | 5 | 5.2491 | 4.1086 | 1.9110 | 0.9512 |
| | | 10 | 6.3197 | 5.3354 | 1.9909 | 0.7431 |
| | | 15 | 5.7394 | 5.0007 | 2.0107 | 0.8376 |
| DAX | Scenario 1 | 5 | 18.6773 | 15.8175 | 9.2827 | 0.8370 |
| | | 10 | 23.4496 | 19.6941 | 11.2345 | 0.7256 |
| | | 15 | 22.8000 | 18.7486 | 11.2457 | 0.7123 |
| | Scenario 2 | 5 | 13.6687 | 12.1355 | 3.7203 | 0.9728 |
| | | 10 | 17.8461 | 15.5167 | 3.7956 | 0.9500 |
| | | 15 | 16.8262 | 14.0575 | 3.7117 | 0.9017 |
| | Scenario 3 | 5 | 11.7707 | 10.4151 | 3.1193 | 0.9770 |
| | | 10 | 15.2011 | 13.1796 | 3.1783 | 0.9581 |
| | | 15 | 14.3498 | 11.9659 | 3.1078 | 0.9179 |
| DJI | Scenario 1 | 5 | 64.1036 | 51.0375 | 4.6789 | 0.7963 |
| | | 10 | 76.7421 | 64.8978 | 4.6717 | 0.6458 |
| | | 15 | 69.6178 | 54.4051 | 4.7296 | 0.7835 |
| | Scenario 2 | 5 | 29.1110 | 24.6261 | 5.7191 | 0.9346 |
| | | 10 | 42.7217 | 37.2901 | 5.5960 | 0.7577 |
| | | 15 | 26.8920 | 22.7033 | 5.6887 | 0.9722 |
| | Scenario 3 | 5 | 25.4506 | 21.5191 | 4.9036 | 0.9439 |
| | | 10 | 37.0988 | 32.3404 | 4.7977 | 0.7923 |
| | | 15 | 23.4998 | 19.8059 | 4.8770 | 0.9761 |
| SSE | Scenario 1 | 5 | 9.3477 | 7.6544 | 3.3557 | 0.8116 |
| | | 10 | 9.7774 | 8.0523 | 3.3336 | 0.7564 |
| | | 15 | 12.6617 | 11.2470 | 3.4131 | 0.6840 |
| | Scenario 2 | 5 | 8.4603 | 7.0721 | 3.2049 | 0.9430 |
| | | 10 | 8.3467 | 7.0678 | 3.1801 | 0.9456 |
| | | 15 | 12.2441 | 11.3312 | 3.3860 | 0.8317 |
| | Scenario 3 | 5 | 7.4032 | 6.1885 | 2.8059 | 0.9501 |
| | | 10 | 7.3038 | 6.1847 | 2.7836 | 0.9524 |
| | | 15 | 10.7141 | 9.9153 | 2.9638 | 0.8528 |

When the results are examined, it is seen that the prediction model, which is based on applying only the decomposition process and no denoising process, has the worst performance among all scenarios. Conversely, the 2LE-CEEMDAN-LSTM-SVR forecasting model emerged as the most effective and successful in comparison.

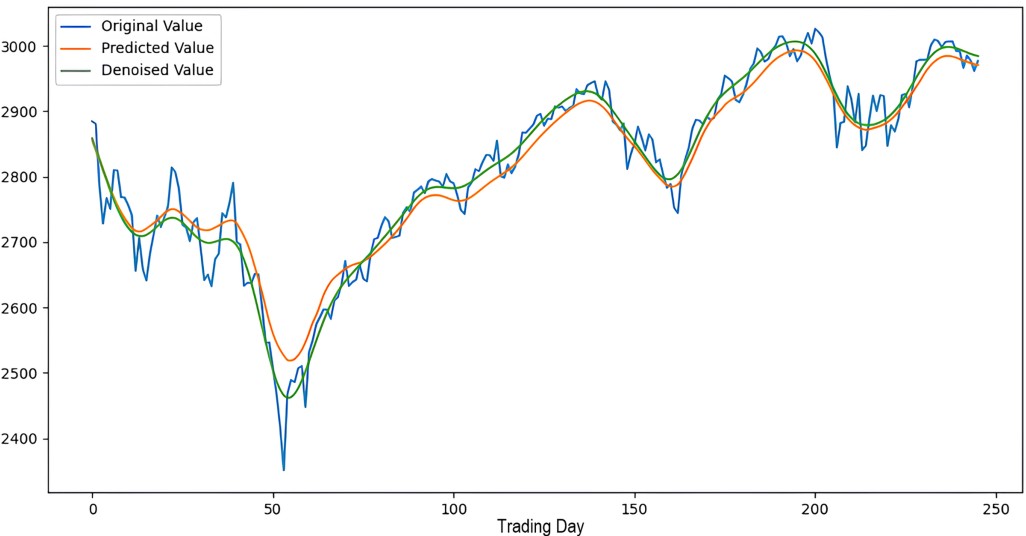

**Figure 5** The success graph of the 2LE-CEEMDAN-LSTM-SVR model on the S&P500 test data.

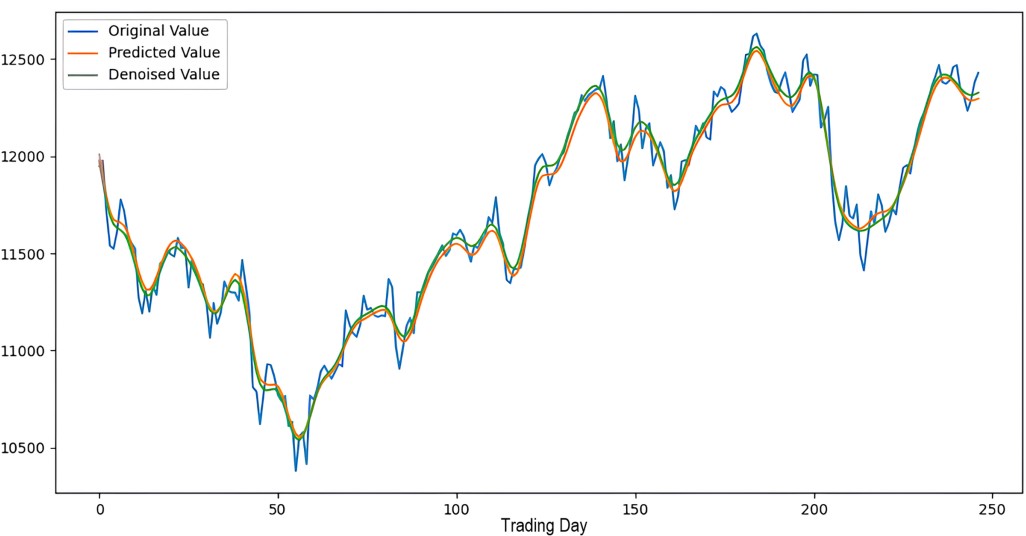

**Figure 6** The success graph of the 2LE-CEEMDAN-LSTM-SVR model on the DAX test data.

Additionally, when comparing the forecasting results performed by considering the past 5, 10, and 15 observation values, it is observed that the forecasts performed using five observation values for S&P500 and DAX, 10 for SSE, and 15 for DJI are more successful. Accordingly, the success graphs of the S&P500, DAX, DJI, and SSE stock market indices in the 2LE-CEEMDAN-LSTM-SVR model are given in Figs. 5–8, respectively. The curve shown in blue in the figures represents the actual closing values of the indices, the green curve shows the denoised data obtained as a result of applying the 2LE-CEEMDAN method, and the orange curve shows the prediction result of the 2LE-CEEMDAN-LSTM-SVR model. When these figures are examined, it is noteworthy that the sharpness in the

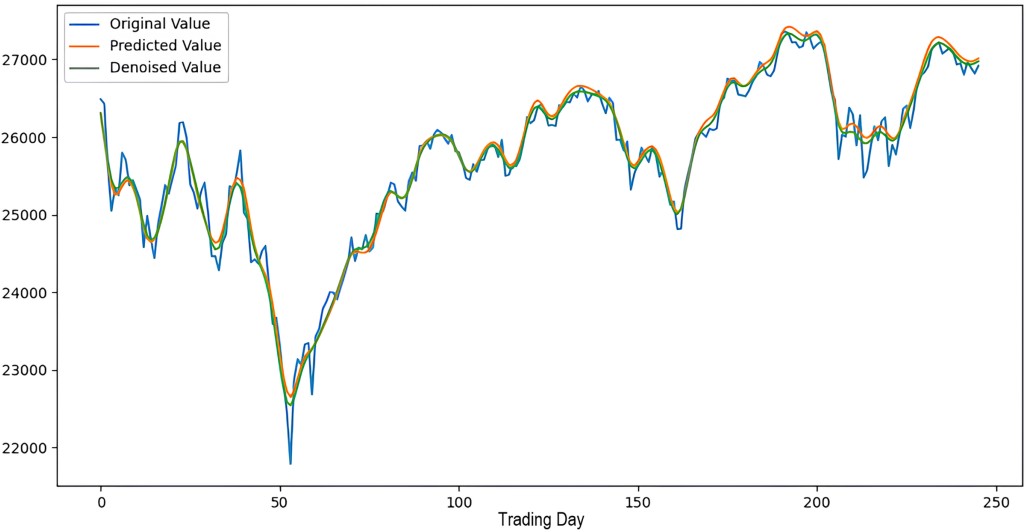

**Figure 7** The success graph of the 2LE-CEEMDAN-LSTM-SVR model on the DJI test data.

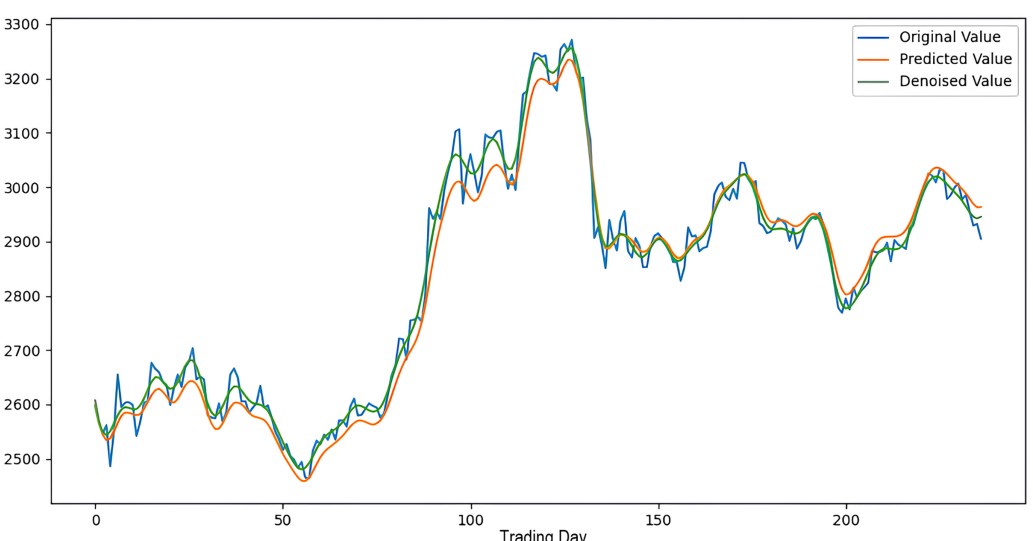

**Figure 8** The success graph of the 2LE-CEEMDAN-LSTM-SVR model on the SSE test data.

closing values of the indices is smoothed by the proposed 2LE-CEEMDAN method, and the forecasting results generally overlap with the denoised data.

## Comparison with other models

In this study, we conducted an experimental comparison among four models to assess the performance of the recommended 2LE-CEEMDAN-LSTM-SVR forecasting model in the context of stock market prediction. Details regarding these models' denoising, decomposition, and forecasting methods are presented in Table 5. Among these models, both CEEMD-CNN-LSTM (*Rezaei, Faaljou & Mansourfar, 2021*) and CEEMDAN-LSTM

**Table 5  Contrastive experiments.**

| Model | Stock market index | Decomposition method | Denoising method | Forecasting model |
|---|---|---|---|---|
| CEEMD-CNN-LSTM (*Rezaei, Faaljou & Mansourfar, 2021*) | S&P500, DJI, DAX, Nikkei225 | Single level CEEMD | – | CNN-LSTM |
| CEEMDAN-LSTM (*Cao, Li & Li, 2019*) | S&P500, DAX, HSI, SSE | Single level CEEMDAN | – | LSTM |
| CAL (*Lv et al., 2022*) | S&P500, DAX, HSI, SSE | Single level CEEMDAN | Augmented Dickey Fuller Test | ARMA-LSTM |
| GRU based on CEEMDAN-Wavelet (*Qi, Ren & Su, 2023*) | S&P500, CSI300 | Single level CEEMDAN | Wavelet transform | GRU |
| 2LE-CEEMDAN-LSTM-SVR | S&P500, DAX, DJI, SSE | Two level CEEMDAN | Entropy ratio-based | LSTM-SVR |

(*Cao, Li & Li, 2019*) did not apply any denoising approach to the IMFs; instead, each IMF was individually trained using the specified forecasting model. Additionally, the CAL (*Lv et al., 2022*) model performed decomposition and denoising processes. This model was applied to the ARMA approach on the IMF, which is more linear as a result of denoising, and the LSTM method was applied to the remaining ones. Finally, the GRU based on CEEMDAN-Wavelet forecasting model (*Qi, Ren & Su, 2023*) used a combination of decomposition and wavelet transformation for denoising financial time series. The wavelet threshold denoising method was applied to the IMFs obtained from the decomposition, and these IMFs were then reconstructed. Subsequently, the reconstructed IMFs were trained using the GRU method.

The 2LE-CEEMDAN-LSTM-SVR forecasting model proposed in our study uses decomposition and denoising methodologies. Diverging from the approaches mentioned in previous studies, our proposed model employs a two-level decomposition coupled with an entropy ratio-based denoising technique. Moreover, in contrast to the strategy of training all noiseless IMFs with a single forecasting model, as observed in the studies by *Rezaei, Faaljou & Mansourfar (2021)* and *Cao, Li & Li (2019)*, our model adopts a differentiated approach. Specifically, the more linear components are trained using the SVR method, while the remaining components are trained using the LSTM method. Additionally, the stock market indices utilized by the models in the comparative analysis are detailed in Table 5.

The performance comparison of the 2LE-CEEMDAN-LSTM-SVR prediction model and the baseline methods proposed for intraday stock market prediction is presented in Table 6. Upon examination of the table, our proposed 2LE-CEEMDAN-LSTM-SVR prediction model demonstrates superior performance compared to the CEEMD-CNN-LSTM and CEEMDAN-LSTM models, which lack the denoising approach and solely rely on the decomposition method. Furthermore, in a comparative analysis of experimental results, it is evident that our 2LE-CEEMDAN-LSTM-SVR model exhibits lower error metrics than prediction models utilizing both denoising and decomposition approaches. This outcome validates the effectiveness of our proposed 2LE-CEEMDAN approach in

**Table 6 Performance comparison.**

| Models | S&P500 | | DAX | | DJI | | SSE | |
|---|---|---|---|---|---|---|---|---|
| | RMSE | MAE | RMSE | MAE | RMSE | MAE | RMSE | MAE |
| CEEMD-CNN-LSTM (*Rezaei, Faaljou & Mansourfar, 2021*) | 13.76 | 10.58 | 84.88 | 65.03 | 155.52 | 118.02 | – | – |
| CEEMDAN-LSTM (*Cao, Li & Li, 2019*) | 4.83 | 3.92 | 33.35 | 24.85 | – | – | 8.74 | 6.86 |
| CAL (*Lv et al., 2022*) | 26.14 | 17.14 | 101.83 | 72.33 | – | – | 19.92 | 14.03 |
| GRU based on CEEMDAN-Wavelet (*Qi, Ren & Su, 2023*) | 22.27 | 17.39 | – | – | – | – | – | – |
| 2LE-CEEMDAN-LSTM-SVR | 5.25 | 4.11 | 11.77 | 10.42 | 23.50 | 19.81 | 7.30 | 6.18 |

reducing noise in financial time series, demonstrating that the 2LE-CEEMDAN-SVR model successfully predicts the closing prices of the next day's stock market indices.

## CONCLUSION AND FUTURE WORKS

This study introduces a novel two-level entropy-based CEEMDAN method, denoted as 2LE-CEEMDAN, to address the noise issue in time series data, specifically applied to the intraday stock market prediction task. The proposed 2LE-CEEMDAN-LSTM-SVR hybrid model integrates frequency decomposition, entropy, LSTM, and SVR. The model comprises two stages: firstly, the 2LE-CEEMDAN method is employed to eliminate noise from the stock market index data, and secondly, the LSTM-SVR prediction model is trained on the obtained denoised data. To assess the effectiveness of the 2LE-CEEMDAN-LSTM-SVR forecasting model, the study applies it to predict the next day's closing value for four major stock market indices: S&P500, DAX, DJI, and SSE. The results demonstrate that the proposed 2LE-CEEMDAN method effectively eliminates noise in financial time series data, positively impacting the model's forecasting performance. Comparative analysis with existing models in the literature reveals that the 2LE-CEEMDAN-LSTM-SVR model outperforms alternative approaches. This study stands out as the first known article to utilize a two-level hierarchical decomposition and an entropy-ratio-based approach to eliminate noise in non-stationary and nonlinear financial time series effectively.

The introduction of a new correlation-based approach for determining IMFs as independent variables in the forecasting model is anticipated to impact the model's predictive performance positively. Additionally, incorporating technical indicators into the model is a potential avenue for further improving its performance. These aspects will be the focus of future work to enhance the robustness and accuracy of the forecasting model.

### Funding

This work was supported by Ondokuz Mayıs University BAP under grant PYO. MUH.1904.23.002. The funders had no role in study design, data collection and analysis, decision to publish, or preparation of the manuscript.

## Grant Disclosures

The following grant information was disclosed by the authors:
Ondokuz Mayıs University BAP: PYO.MUH.1904.23.002.

## Competing Interests

The authors declare that they have no competing interests.

## Author Contributions

- Zinnet Duygu Akşehir conceived and designed the experiments, performed the experiments, analyzed the data, performed the computation work, prepared figures and/or tables, authored or reviewed drafts of the article, and approved the final draft.
- Erdal Kılıç conceived and designed the experiments, analyzed the data, authored or reviewed drafts of the article, and approved the final draft.

## Data Availability

The data and code is available at GitHub and Zenodo:
- https://github.com/daksehir/2LE-CEEMDAN-LSTM-SVR.git.
- daksehir. (2023). daksehir/2LE-CEEMDAN-LSTM-SVR: 2LE-CEEMDAN-LSTM-SVR (0.1.0). Zenodo. https://doi.org/10.5281/zenodo.10438696.

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
