# Peer review of "A new denoising approach based on mode decomposition applied to the stock market time series: 2LE-CEEMDAN"

_PeerJ Computer Science, doi:10.7717/peerj-cs.1852_

## Round 0.1 · original submission · Major Revisions

With reviewers comments add some resent work. I am also suggest to explain your results with more details

**Language Note:** The review process has identified that the English language must be improved. PeerJ can provide language editing services - please contact us at copyediting@peerj.com for pricing (be sure to provide your manuscript number and title). Alternatively, you should make your own arrangements to improve the language quality and provide details in your response letter. – PeerJ Staff

Reviewer 1 ·

Basic reporting

The author proposed a model called "A new denoising approach based on mode decomposition applied to the stock market time series: 2LE-CEEMDAN" The topic is interesting. However, there are a lot of shortcomings which need to be addressed, which are;
1. The authors need to rewrite the article in a completely different manner, concentrating on explaining their work's significance and contributions, which are very important to mention.
2. The data sets and parameters are not clearly set and mentioned in the proposed research.
3. The figures are very low quality; they need to be improved to better quality.
4. What is the novelty of the paper?
5. State-of-the-art literature is missing; the author needs to en-cooperate with the latest related work
6. There are many typos and grammatical mistakes that need to be addressed in revision.

Experimental design

Comments are in the main report.

Validity of the findings

Comments are in the main report.

Additional comments

No comments

Reviewer 2 ·

Basic reporting

1) Split the literature review and introduction into separate sections. Move Motivation, Contribution, and Paper Organization to the Introduction.
2) The current literature review is too limited. Include a broader range of papers.
3) Figures lack x-axis descriptions. Please add these details.
4) Kindly elaborate on your model's complexity and predictions. Explain the tables in the comparison analysis and include a cost analysis in performance evaluation.
5) I would suggest providing a detailed discussion on the hyperparameter tuning process for the LSTM and SVR models.
6) The source code was not found in the provided materials in the source files. Please include it for result transparency.
7) Add an analysis of the computation time and compare it with the previous schemes.
8) The current references are insufficient, with limited recent papers. Include more papers from 2023 and 2022 for up-to-date context. Currently, there is only one paper from 2023 and 4 papers from 2022.

Experimental design

All the comments are added in the basic reporting

Validity of the findings

All the comments are added in the basic reporting

Additional comments

All the comments are added in the basic reporting

---

## Round 0.2 · accepted · Accept

Based on the reviewer recommendation the paper is now acceptable.

Reviewer 1 ·

Basic reporting

The author addressed all comments, so I decided to accept the paper in its current form.

Experimental design

Ok

Validity of the findings

Enough

Additional comments

Nil

---

## Author Rebuttal · Round 0.2

Dear Editors,

We sincerely thank the respected reviewers for their invaluable comments and thoughtful feedback on the manuscript. Their meticulous evaluation has been instrumental in refining the quality and clarity of our work.

Thank you for the opportunity to address the reviewers' comments and enhance the manuscript. We believe that the revisions have substantially improved the quality of the work, making it well-suited for publication in PeerJ.

Zinnet Duygu Akşehir

On behalf of all authors.

**Editor Comments:**

\*\*Language Note:\*\* The review process has identified that the English language must be improved. PeerJ can provide language editing services - please contact us at copyediting@peerj.com for pricing (be sure to provide your manuscript number and title). Alternatively, you should make your own arrangements to improve the language quality and provide details in your response letter. – PeerJ Staff

The article has been rewritten, and it has undergone a language review by a colleague who is proficient in English.

**Reviewer 1:**

*Basic Reporting*

*Reviewer #1, Concern #1: The authors need to rewrite the article in a completely different manner, concentrating on explaining their work's significance and contributions, which are very important to mention.*

The manuscript has been thoroughly rewritten to address the highlighted issues and concerns. The revisions aim to enhance the clarity and presentation of the research, ensuring a more effective communication of the work's significance and contributions.

*Reviewer #1, Concern #2: The data sets and parameters are not clearly set and mentioned in the proposed research.*

The dataset utilized in the study was elaborated on as a subsection within the 'Experimental Settings' section. This subsection covered information on the indices used, the date ranges for the considered closing prices, the data collection sources, and descriptive statistics related to the dataset. In response to the reviewer's comment (pg 11, lines 333-344), additional detailed information has been provided concerning the stock market indices employed in the study.

Furthermore, the construction of the forecasting model is detailed in the subsequent subsection, including parameters associated with the prediction models. Specifics regarding the hyperparameter settings of the models are presented on page 12, lines 376-382.

*Reviewer #1, Concern #3: The figures are very low quality; they need to be improved to better quality.*

While submitting the manuscript, the figures within the study were simultaneously checked for image quality. The existing figures have successfully passed this quality check.

*Reviewer #1, Concern #4: What is the novelty of the paper?*

The innovative aspects of the study are elaborated in detail within the Motivation and Contributions subsection (pg 3-4, line 136-154). Furthermore, the significance and innovative contributions of the study are consistently underscored throughout the manuscript.

In this study, a new approach that effectively separates noisy data from financial time series was developed without causing any loss of useful information. Therefore, this aspect distinguishes this study from existing literature, emphasizing a novel perspective.

The contributions of the study, along with its superior and innovative aspects in comparison to existing studies in the literature, are outlined below:

1. A novel denoising method, named 2LE-CEEMDAN, has been developed based on CEEMDAN to extract valuable information from high-frequency components discarded as noise.
2. A new methodology has been proposed to effectively identify noisy components from the IMFs obtained through the decomposition of time series data by utilizing approximate and sample entropy to measure irregularities in the time series data.
3. To demonstrate the method's effectiveness, a new hybrid prediction model has been presented for predicting the closing prices of stock market indices. This hybrid model includes the 2LE-CEEMDAN denoising approach with LSTM and SVR methods.

*Reviewer #1, Concern #5: State-of-the-art literature is missing; the author needs to en-cooperate with the latest related work*

The literature review section of this study was expanded by examining a total of 6 current studies in this context in the literature, including five from 2023 and one from 2022. These articles are:

1. Qi, C., Ren, J., & Su, J. (2023). GRU Neural Network Based on CEEMDAN–Wavelet for Stock Price Prediction. *Applied Sciences*, *13*(12), 7104.
2. Wang, J., Cheng, Q., & Dong, Y. (2023). An XGBoost-based multivariate deep learning framework for stock index futures price forecasting. *Kybernetes*, *52*(10), 4158-4177.
3. Cui, C., Wang, P., Li, Y., & Zhang, Y. (2023). McVCsB: A new hybrid deep learning network for stock index prediction. *Expert Systems with Applications*, 120902.
4. Zhao, Y., & Yang, G. (2023). Deep Learning-based Integrated Framework for stock price movement prediction. *Applied Soft Computing*, *133*, 109921.
5. Roostaee, M. R., & Abin, A. A. (2023). Forecasting financial signal for automated trading: An interpretable approach. *Expert Systems with Applications*, *211*, 118570.
6. Rekha, K. S., & Sabu, M. K. (2022). A cooperative deep learning model for stock market prediction using deep autoencoder and sentiment analysis. *PeerJ Computer Science*, *8*, e1158.

Additionally, a comparison is made with the study mentioned as number 1, which employs both decomposition and denoising approaches for predicting the closing price of various stock market indices. The comparison results revealed that the 2LE-CEEMDAN-LSTM-SVR model proposed in our

study achieved more accurate predictions. Additionally, our suggested 2LE-CEEMDAN method effectively separated noise from the index data.

The "Comparison with Other Models" subsection (pg 16-18) provides details about the comparison.

*Reviewer #1, Concern #6: There are many typos and grammatical mistakes that need to be addressed in revision.*

In response to the first comment, efforts were made to rectify the pertinent grammatical and typos mistakes during the rewritten of the study.

**Reviewer 2:**

*Basic Reporting*

Reviewer #2, Concern #1: Split the literature review and introduction into separate sections. Move Motivation, Contribution, and Paper Organization to the Introduction.

The Introduction section is divided into three subsections: Related Work, Motivation and Contributions, and Organization.

Reviewer #2, Concern #2: The current literature review is too limited. Include a broader range of papers.

The literature review section has been expanded by adding six recent studies, with five published in 2023 and one in 2022. These articles are:

1. Qi, C., Ren, J., & Su, J. (2023). GRU Neural Network Based on CEEMDAN–Wavelet for Stock Price Prediction. *Applied Sciences*, *13*(12), 7104.
2. Wang, J., Cheng, Q., & Dong, Y. (2023). An XGBoost-based multivariate deep learning framework for stock index futures price forecasting. *Kybernetes*, *52*(10), 4158-4177.
3. Cui, C., Wang, P., Li, Y., & Zhang, Y. (2023). McVCsB: A new hybrid deep learning network for stock index prediction. *Expert Systems with Applications*, 120902.
4. Zhao, Y., & Yang, G. (2023). Deep Learning-based Integrated Framework for stock price movement prediction. *Applied Soft Computing*, *133*, 109921.
5. Roostaee, M. R., & Abin, A. A. (2023). Forecasting financial signal for automated trading: An interpretable approach. *Expert Systems with Applications*, *211*, 118570.
6. Rekha, K. S., & Sabu, M. K. (2022). A cooperative deep learning model for stock market prediction using deep autoencoder and sentiment analysis. *PeerJ Computer Science*, *8*, e1158

Reviewer #2, Concern #3: Figures lack x-axis descriptions. Please add these details.

The expression "trading day" was located on the x-axis of the figures in the submitted manuscript. However, based on this comment, the "trading day" text was made more obvious.

Reviewer #2, Concern #4: Kindly elaborate on your model's complexity and predictions. Explain the tables in the comparison analysis and include a cost analysis in performance evaluation.

The comparison with existing studies in the literature was detailed and the following statements were added to "comparison with other models" subsection:

"The performance comparison of the 2LE-CEEMDAN-LSTM-SVR prediction model and the baseline methods proposed for intraday stock market prediction is presented in Table 6. Upon examination of the table, our proposed 2LE-CEEMDAN-LSTM-SVR prediction model demonstrates superior

performance compared to the CEEMD-CNN-LSTM and CEEMDAN-LSTM models, which lack the denoising approach and solely rely on the decomposition method. Furthermore, in a comparative analysis of experimental results, it is evident that our 2LE-CEEMDAN-LSTM-SVR model exhibits lower error metrics than prediction models utilizing both denoising and decomposition approaches. This outcome validates the effectiveness of our proposed 2LE-CEEMDAN approach in reducing noise in financial time series, demonstrating that the 2LE-CEEMDAN-LSTM-SVR model successfully predicts the closing prices of the next day's stock market indices."

In the comparison analysis, we observed that three of the four studies under consideration employed the same LSTM architecture in our proposed model. The remaining study introduced the GRU architecture; however, details about this architecture were not provided. It is crucial to highlight that our approach diverges from the other four studies by incorporating a two-level decomposition process. If we had opted for a single decomposition step, the complexity of our model would have been similar to the four compared models. However, due to our two-level decomposition approach, the complexity of our proposed prediction model is slightly higher than that of other models. Additionally, since we train the prediction models offline, the increasing complexity of our model does not pose any problem.

Reviewer #2, Concern #5: I would suggest providing a detailed discussion on the hyperparameter tuning process for the LSTM and SVR models.

The hyperparameter settings for the LSTM and SVR prediction models are detailed in Table 2. The hyperparameter tuning process for the SVR method utilized a grid search approach. Hyperparameter values for LSTM were chosen based on relevant literature studies, incorporating commonly preferred values in this study. Experiments were executed by generating training-test sets concerning three different values for the time step parameter specified in the table. Moreover, to mitigate overfitting during training in the LSTM method, a dropout layer with a ratio of 0.1 was introduced between both hidden layers. Additionally, early stopping was implemented.

These details about the hyperparameter tuning process have been added to the last paragraph of the "Construction of forecasting model" subsection (pg 12, line 376-382).

Reviewer #2, Concern #6: The source code was not found in the provided materials in the source files. Please include it for result transparency.

The GitHub link to the source code was included during the manuscript upload. The code is accessible through the following link:

https://github.com/daksehir/2LE-CEEMDAN-LSTM-SVR

Reviewer #2, Concern #7: Add an analysis of the computation time and compare it with the previous schemes.

Computation time can vary considering factors such as input data used, the choice of decomposition algorithm, the number of parameters in the model, and the hardware on which the model is executed. In the case where a similar architectural structure and input data are employed and the model is run on the same hardware, similar computation time may be observed across the four studies under consideration. However, it is important to note that computation time has not been explicitly calculated or reported in any of the works in the context of stock market prediction studies. Consequently, since there is no data available regarding computation time in these studies, making a meaningful comparison is not feasible.

Reviewer #2, Concern #8: The current references are insufficient, with limited recent papers. Include more papers from 2023 and 2022 for up-to-date context. Currently, there is only one paper from 2023 and 4 papers from 2022.

The literature review section of this study was expanded by examining a total of 6 current studies in this context in the literature, including five from 2023 and one from 2022. These articles are:

1. Qi, C., Ren, J., & Su, J. (2023). GRU Neural Network Based on CEEMDAN–Wavelet for Stock Price Prediction. *Applied Sciences*, *13*(12), 7104.
2. Wang, J., Cheng, Q., & Dong, Y. (2023). An XGBoost-based multivariate deep learning framework for stock index futures price forecasting. *Kybernetes*, *52*(10), 4158-4177.
3. Cui, C., Wang, P., Li, Y., & Zhang, Y. (2023). McVCsB: A new hybrid deep learning network for stock index prediction. *Expert Systems with Applications*, 120902.
4. Zhao, Y., & Yang, G. (2023). Deep Learning-based Integrated Framework for stock price movement prediction. *Applied Soft Computing*, *133*, 109921.
5. Roostaee, M. R., & Abin, A. A. (2023). Forecasting financial signal for automated trading: An interpretable approach. *Expert Systems with Applications*, *211*, 118570.
6. Rekha, K. S., & Sabu, M. K. (2022). A cooperative deep learning model for stock market prediction using deep autoencoder and sentiment analysis. *PeerJ Computer Science*, *8*, e1158.

Additionally, a comparison is made with the study mentioned as number 1, which employs both decomposition and denoising approaches for predicting the closing price of various stock market indices. The comparison results revealed that the 2LE-CEEMDAN-LSTM-SVR model proposed in our study achieved more accurate predictions. Additionally, our suggested 2LE-CEEMDAN method effectively separated noise from the index data.

The "Comparison with Other Models" subsection (pg 16-18) provides details about the comparison.